# Chaos Based Cryptographic Pseudo-Random Number Generator Template with Dynamic State Change †

**Octaviana Datcu** *, **Corina Macovei** and **Radu Hobincu**

University POLITEHNICA of Bucharest, Faculty of Electronics, Telecommunications and Information Technology, Bd. Iuliu Maniu 1-3, 061071 Bucharest, Romania; corina.macovei@stud.etti.upb.ro (C.M.); radu.hobincu@upb.ro (R.H.)

* Correspondence: octaviana.datcu@upb.ro
† This paper is an extended version of our paper published in The 42nd International Conference on Telecommunications and Signal Processing (TSP), Budapest, Hungary, 1–3 July 2019.



**Featured Application: The application implemented for the proposed pseudo-random number generator is a one-time pad stream cipher that is able to encrypt an 8 K PNG RGB color image in 2 s.**

**Abstract:** This article presents a configurable, high-throughput pseudo-random number generator template targeting cryptographic applications. The template is parameterized using a chaotic map that generates data, an entropy builder that is used to periodically change the parameters of the map and a parameter change interval, which is the number of iterations after which the entropy builder will change the generator's parameters. The system is implemented in C++ and evaluated using the TestU01 and NIST RNG statistical tests. The same implementation is used for a stream cipher that can encrypt and decrypt PNG images. A Monte-Carlo analysis of the seed space was performed. Results show that for certain combinations of maps and entropy builders, more than 90% of initial states (seeds) tested pass all statistical randomness tests. Also, the throughput is large enough so that a 8 K color image can be encrypted in 2 s on a modern laptop CPU (exact specifications are given in the paper). The conclusion is that chaotic maps can be successfully used as a building block for cryptographic random number generators.

**Keywords:** chaos-based pRNGs; randomness tests; telecommunications; information systems; digital signal processing; image signal processing

## 1. Introduction

As bandwidth in state-of-the-art communication channels is increasing at a high rate, cryptographic solutions need to keep up with the large amount of information that should be encrypted. One of the key requirements for stream cryptosystems is the underlying pseudo-random number generator. In recent years, chaotic systems have been considered as generating good and fast pseudo-random sequences [1–11]. The generated numbers are equiprobable, non-correlated. Good pseudo-random number generators (pRNGs) have a large period of repetition. They have properties such as reproducibility and consistency (independence from the seed), portability, efficiency, coverage of the entire output space. The sequences generated utilizing different seeds must be disjoint, consecutive numbers not revealing any pattern for any length of the sequence. Every permutation of a number generated by a good pRNG is equally likely [12]. The theory behind a good pRNG and some practical aspects concerning its design can be consulted in [13]. An analysis of statistical testing methods of pRNGs is given in [14].

Generally, cryptographically secure pseudo-random sequences are obtained from cryptosystems—stream ciphers or block ciphers in counter mode—which, as opposed to regular pRNGs, are slower [15]. In this paper we attempt to design and build a chaos based pseudo random number generator template, configurable with a map function and entropy builder so that depending on the application, it can focus on randomness quality, throughput, or a balance between the two.

A chaos based pseudo-random number generator was proposed in [16]. The underlying system was the three dimensional generalized Hénon map (2). Given its simple shift-register form, yet complex behavior, many studies were done [17–25] to investigate its dynamics.

$$
\begin{aligned}
x_{k+1} &= a - y_k^2 - bz_k \\
y_{k+1} &= x_k \\
z_{k+1} &= y_k
\end{aligned}
\tag{1}
$$

where $a \in (0,2)$, $b \in [-0.3, 0.3]$, $x, y, z \in (-2, 2)$, $k$ is the iteration number and $(x_0, y_0, z_0)$ the initial states. The definition intervals for parameters $a$ and $b$ were established by computing bifurcation diagrams [26] and Lyapunov exponents [27] for $a \in (-2, 2)$ and $b \in (-1, 1)$. A Monte Carlo analysis [28] with initial conditions $x_0, y_0, z_0$ chosen randomly from an uniform distribution in $[-2,2]$ was performed. The conclusion was that $a \in (0,2)$ and $b \in [-0.3, 0.3]$ are the ranges ensuring $x, y, z \in (-2, 2)$.

For the pRNG in [16] the least significant nibble of each state of system (2) was discarded and the next bytes *xor-ized*, as shown in Figure 1. A sequence obtained was tested for randomness with the NIST [14] battery test.

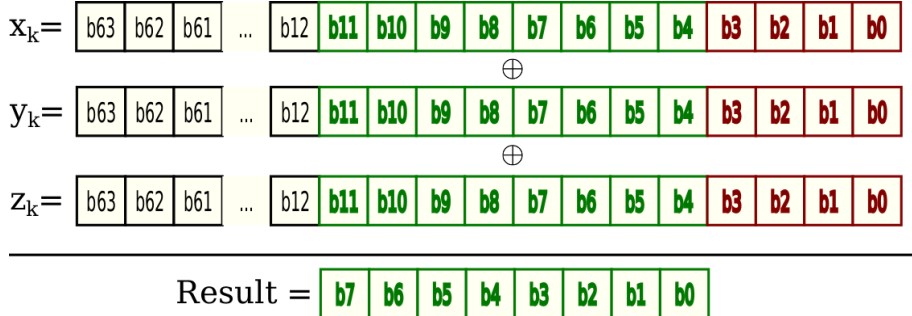

**Figure 1.** Logic for generating random bytes from the states' double precision floating point strings.

As the system of equations generating the pseudo-random sequences is chaotic—aperiodic and highly sensitive to initial states—it displays a fractal structure [29], depending upon its bifurcation parameters, $(a; b)$. This implies that not all pairs of parameters engender a pseudo-random behavior for the generated bits and also that there is no continuous interval that contain only chaotic parameter values [30]. Different evolutions are depicted in Figure 2 for the same initial conditions:

$$(x_0, y_0, z_0) = (0.814723686393179, 0.905791937075619, 0.126986816293506)$$

and different pairs $(a; b)$ given in Table 1. The top left and the bottom left state spaces show periodic attracting orbits, while the right part of the figure demonstrates (hyper)chaotic attractors.

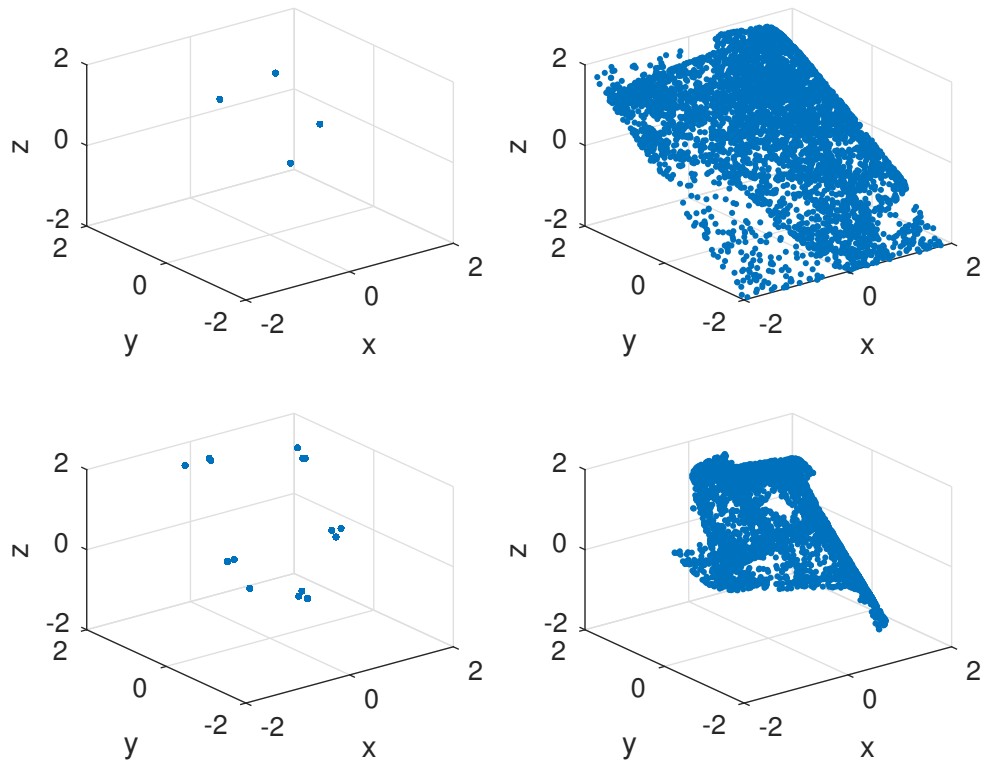

**Figure 2.** Phase portraits of the generalized three-dimensional Hénon map for different $(a; b)$ pairs.

**Table 1.** The parameter values for the system evolutions displayed in Figure 2.

| Position | a | b |
|---|---|---|
| top left | 1.066073952077004 | −0.122171054133312 |
| bottom left | 1.822849696327467 | −0.210260146847663 |
| top right | 1.422086796614167 | −0.292608160780357 |
| bottom right | 1.566451149781642 | 0.210678068488918 |

The Lyapunov exponents [27] are the commonly used metric to indicate periodic or (hyper)chaotic behavior. For the investigated system, the generalized Hénon map (2), the number of exponents is three, corresponding to the dimension of the state space. The algorithm computing the Lyapunov spectrum in [27] sorts the exponents from the largest to the lowest. When the system generates rapidly divergent trajectories, the largest exponent, $\lambda_1$, is positive, indicating random-like (chaotic) behavior. A positive value for the second Lyapunov exponent also is the mark of a hyperchaotic evolution, random-like dynamics in two directions of the space. The last exponent has to be negative to maintain the bounded space. Figure 3 shows the three Lyapunov exponents for a fixed value of the parameter $b$ and varying $a$. Initial conditions, although irrelevant for this metric, $(x_0, y_0, z_0)$, are the same as above.

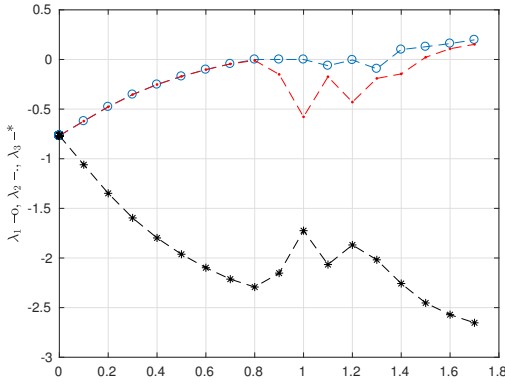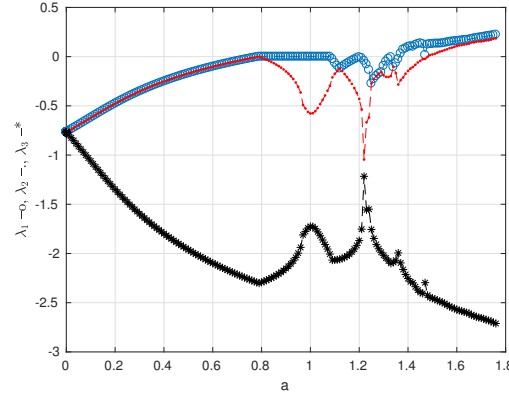

**Figure 3.** Lapunov exponents for $b = 0.1$ and varying $a$ in $[0;2)$ using different computation steps. $\lambda_1$ in blue, $\lambda_2$ in red, $\lambda_3$ in black.

In [24] the authors also attempted an FPGA implementation for the generator, comparing the results with a highly optimized FPGA implementation of a cellular automaton pRNG in [31]. The conclusion was that due to the recursive nature of the chaotic functions and the fact that the computation must be done in double precision floating point, the performance of the design will be limited, hardware-specific optimizations like pipelining and speculative execution being unfeasible. This makes the design FPGA-resistant which in certain types of applications, like blockchain, may even be an advantage.

As already stated, the values for $\lambda_1$ should be positive. When setting $b = 0.1$, the only interval for $a$ where the pseudo-random behavior is enabled by the 3D generalized Hénon map is $[1.3; 1.8)$. The two coefficients are highlighted in Figure 4. Different values for the step used to compute their values (top left: $10^{-2}$, top right: $10^{-3}$, bottom left: $10^{-4}$, bottom right: $10^{-5}$) reveal more and more undesired values for the pair $(a, 0.1)$—as the step is smaller. Those pairs are the ones for which $\lambda_1 < 0$. The fact that there are no continuous intervals of values for either $a$ or $b$ for which the map is chaotic or hyper-chaotic makes it difficult to choose or to validate values for the parameters and the initial states such that the generator does not enter a periodic loop. Since these five values represent the *seed* of the generator, it is crucial that their choice is appropriate. In [32] the solutions presented were either to define a system that will transform an invalid seed into a correct one during the system's initialization, or to periodically alter the system's trajectory during runtime, in order to avoid stable orbits.

The first research direction, towards seed validation, concluded with [23] and [33], where it was shown that Lyapunov exponents could be a solution for determining if a tuple of initial states and parameter values are suitable for randomness generation. However, there are still a large number of operations required for computing the exponents, which introduces an unwanted latency to the system's initialization. Also, while there is an obvious correlation between the chaotic behavior of the map and randomness quality of the generated sequence, some inconsistencies were observed: although the Lyapunov exponents were positive, the sequence failed to pass the NIST tests and vice-versa.

In the end, the focus turned towards the second approach, where a perturbation is introduced periodically in the evolution of the system to circumvent stable orbits, the initial research being published in [32]. The new research in this paper adds the following elements over the previous one: details the simulation step for the bifurcation diagrams and Lyapunov exponents to reveal the stable orbit issue more clearly in Section 1, introduces a new chaotic map template and new entropy builders and develops the architecture of a configurable generator design for a wide range of applications in Section 2, tests the design with an additional battery (TestU01), performs a throughput analysis and compares the results with three different types of generators in Section 3, and finally, integrates the system in a working application—a one-time pad stream cryptosystem for ultra high definition images in Section 4. The code for the application is available on https://gitlab.dcae.pub.ro/research/chaos/

AppliedSciencesPaper. Section 5 concludes this paper with a discussion of the results and a proposal for future development.

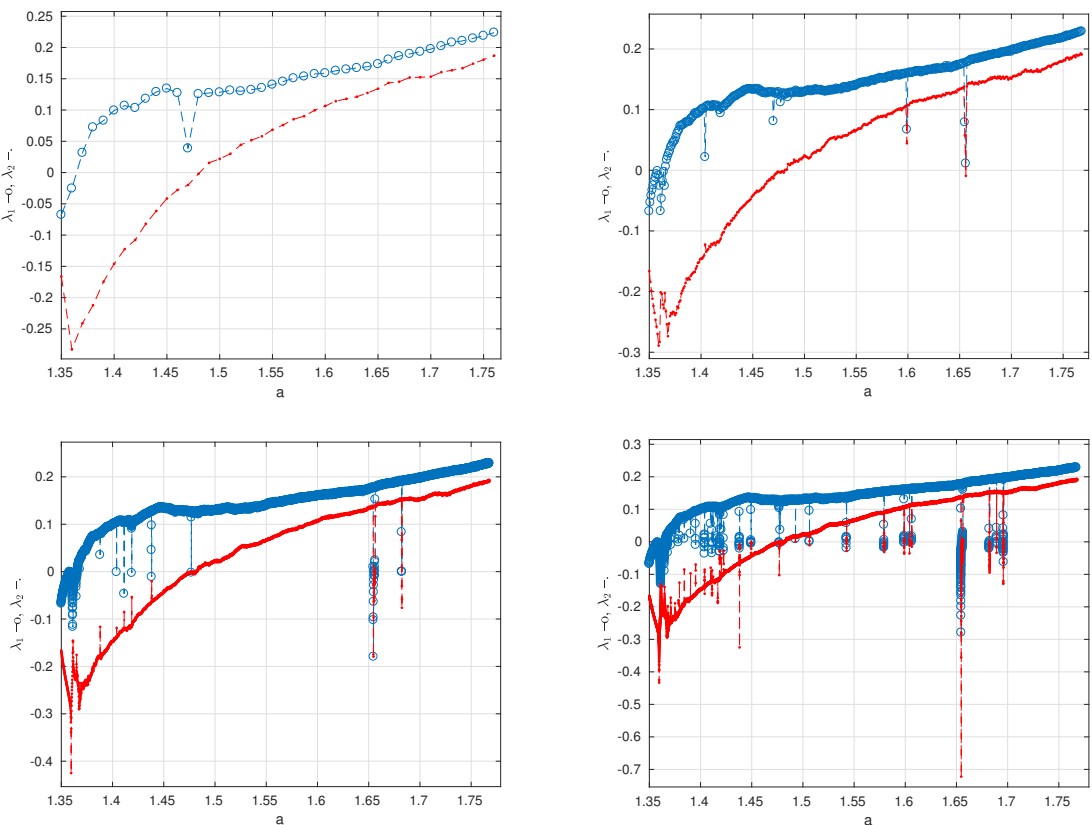

**Figure 4.** The largest Lapunov exponents for $b = 0.1$, $a \in [1.35; 1.8]$. $\lambda_1$ in blue, $\lambda_2$ in red, for different computation steps.

## 2. New Generator Design

As a modification to the first version of the generator [16] which only used one chaotic map, the generalized Hénon map, and never altered the internal states other than by simply computing the map's formula, the second version [32] added a block that accumulates the values of all intermediate states, gathering entropy. This block was named *entropy builder*. Once every $n$ iterations (where $n$ is a parameter of the generator called the *parameter change interval* in the rest of the paper) the accumulator was used to change the map's parameters according to (2).

In the current paper we introduce the third version of the generator along with a C++ implementation, where the generator map as well as the entropy builder are configurable. An advantage of this approach is that the system is modular and flexible, allowing configuration with any chaotic map and with several algorithms for building entropy. Also, selecting particular configurations will yield high throughput performance, as other configurations will increase the quality of the random sequence. The block diagram is presented in Figure 5 where the green blocks are parameterized and the byte generator block is depicted in Figure 1.

$$
\begin{aligned}
a &= E \mod 2.0 \\
b &= E \mod 0.6 - 0.3
\end{aligned}
\tag{2}
$$

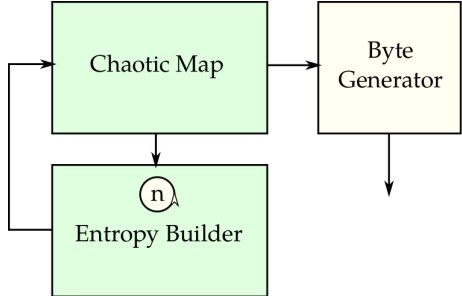

**Figure 5.** Block diagram of the new system proposal. Green blocks are configurable.

Each chaotic map (see Figure 5) has a number of parameters, $||P||$, and a number of states $||S||$. As an example, the 3D generalized Hénon has two parameters denoted by $P = (a; b)$ and three states denoted by $S = (x; y; z)$. Every iteration, the entropy builder takes tuples of $||S||$ values (the current state values) and accumulates entropy. Every $n$ iterations it also provides a tuple of $||P||$ values that replaces the current parameter set of the map. The entropy builder and the chaotic map have to match the same $||P||$ and $||S||$ to obtain a valid configuration. A general form for a three-dimensional map is given in (4).

$$\begin{aligned} x^+ &= y \\ y^+ &= z \\ z^+ &= f(x, y, z) \end{aligned} \tag{3}$$

The maps tested in this paper include the 3D Hénon and all maps presented in [34] and given in Table 2. The formula in row 13 is the generic form of the chaotic map used.

As entropy builders, besides the initial sum approach, named $E_1$, one more design was added, $E_2$, both given in (4). Since there are two parameters involved in the computation of the 3D generalized Hénon map, a third approach was to use a combination of both $E_1$ and $E_2$, one for each of the parameters, with good results.

$$\begin{aligned} E_1 &= E_1 + x_k + y_k + z_k \\ E_2 &= E_2 + sin((z_k - y_k)/2) \end{aligned} \tag{4}$$

**Table 2.** Nonlinear functions [34].

| ID | Maps | Cases |
|----|------|-------|
| 1 | $x_k - x_k \cdot y_k + x_k \cdot z_k + 0.5$ | $NFI_1$ |
| 2 | $x_k + x_k^2 + z_k^2 - 2 \cdot x_k \cdot y_k - 0.2$ | $NFI_2$ |
| 3 | $1.7 \cdot x_k - 0.7 \cdot z_k + z_k^2 - x_k \cdot y_k - 0.3$ | $NFI_3$ |
| 4 | $1.07 \cdot x_k - 0.07 \cdot z_k - x_k \cdot y_k + x_k \cdot z_k + 0.28$ | $NFI_4$ |
| 5 | $x_k + 0.2 \cdot x_k^2 + 0.8 \cdot y_k^2 - x_k \cdot z_k - 0.1$ | $NFI_5$ |
| 6 | $x_k + 0.19 \cdot x_k^2 - 0.22 \cdot z_k^2 + 0.03 \cdot y_k \cdot z_k + 0.03$ | $NFI_6$ |
| 7 | $-z_k + x_k \cdot y_k + y_k \cdot z_k + 1$ | $NFII_1$ |
| 8 | $x_k - 0.2 \cdot z_k^2 + 0.3 \cdot x_k \cdot y_k + 1$ | $NFII_2$ |
| 9 | $-z_k^2 + 0.7 \cdot x_k \cdot y_k + x_k \cdot z_k + 0.7$ | $NFII_3$ |
| 10 | $-y_k - 0.1 \cdot x_k \cdot y_k + x_k \cdot z_k + 1.2$ | $NFII_4$ |
| 11 | $0.9 \cdot x_k - z_k + 0.7 \cdot x_k \cdot y_k + 0.9$ | $NFII_5$ |
| 12 | $0.35 \cdot z_k^2 - 0.57 \cdot x_k \cdot y_k - 1.27$ | $NFII_6$ |
| 13 | $a_0 x_k + a_1 y_k + a_2 z_k + a_3 x_k^2 + a_4 y_k^2 + a_5 z_k^2 + a_6 x_k y_k + a_7 x_k z_k + a_8 y_k z_k + a_9$ | $NF\_GEN$ |

## 3. Generator Implementation and Testing

### 3.1. Testing Setup

The experimental implementation done in C++ defines a *GeneratorTemplate* template class that can be parameterized with a specific map function and an entropy builder. Four applications were developed for testing and evaluation:

1.  *generator*—an application that takes as arguments a byte count, the name of the map, the entropy builder, the number of iterations for time spacing and the initial state values and parameters and generates a text file with the specified number of bytes, one per line, in binary format;
2.  *test_suite*—an application that takes as arguments the name of the map, the entropy builder, the number of iterations for time spacing and the initial state values and parameters and runs the *TestU01* and *NIST* test batteries, outputting the results in a corresponding file, and appending a summary to the results.csv file;
3.  *throughput_test*—an application that takes as arguments a byte count, the name of the map, the entropy builder, the number of iterations for time spacing and the initial state values and parameters and computes the throughput of the specified generator in MB/s;
4.  *codec*—an application that takes as arguments the name of the input PNG file, the name of the output PNG file, the name of the map, the entropy builder, the number of iterations for time spacing and the initial state values and parameters and encrypts (or decrypts, as the stream cipher is symmetric) the input image, writing the results in the output file.

The *generator* can be used to write pseudo-random data to a file for later use in other third party applications or test batteries, like *Diehard* [35]. For this section however, the relevant executables are *test_suite* and *throughput_test*.

The randomness statistical tests used to validate the generator are *TestU01* [36] running either *SmallCrush*, *Crush* or *BigCrush* suites and the *NIST* [14] test battery. *TestU01* is installed in the system and dynamically linked to the testing application as opposed to the *NIST* suite which is integrated at code level. The *NIST* suite has also suffered minor code changes in order to strip down the test harness and run the tests directly. *TestU01* is called using its own API which provides a simple and direct way of specifying a function to call in order to receive a new pseudo-random word. *NIST* however required some additional work since all the tests are designed to read pseudo-random data from a file, so the testbench starts by generating all required data and writing it into a file and then calling *NIST*'s *asses* function which in turn will invoke each test that will read pseudo-random data from the aforementioned file. NIST is configured to test 12 streams of 1 million bytes for each tested seed value. *TestU01*'s *SmallCrush* is used in all tests presented in this paper. The entire testbench design is shown in Figure 6.

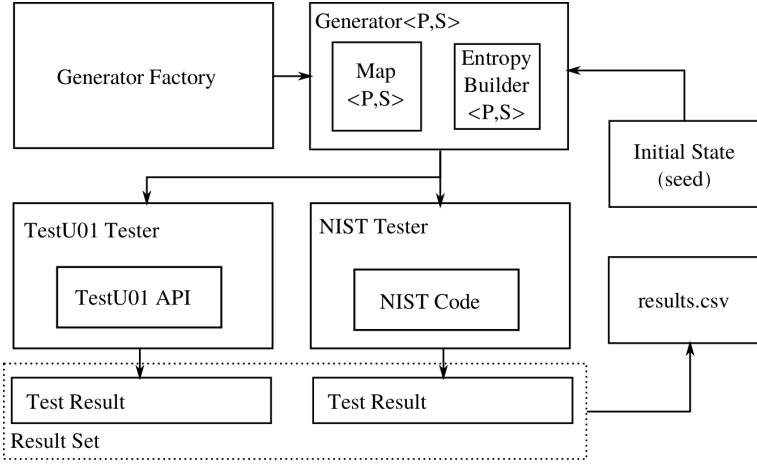

**Figure 6.** Testbench block diagram.

As a comparison with state-of-the art pRNGs, three algorithms were selected and tested along with the current system proposal:

1. **Mersenne Twister**—a non-cryptographically secure pRNG, described in [37] and implemented in the GCC C++ standard library;
2. **ChaCha**—a cryptographically secure pRNG, from the stream cipher implementation in [38];
3. **Adaptive Chirikov Map pRNG**—another chaos-based pRNG proposed in [9] which uses two adaptive Chirikov maps and a comparison procedure to generate random bits.

The following subsection presents the results of three experiments done in order to qualify the generator. First, a visual evaluation for the $(a, b)$ pairs generated by each entropy builder algorithm was made by plotting the pairs in a 2D space. The expectation is to see better randomness from plots that cover the entire plane as opposed to those showing distinctive patterns.

The second experiment chose certain tuples of parameters and initial states that are known to produce periodic trajectories of the considered map. The experiment determines if such seeds can still be used by generators if a suitable entropy builder is used to break the patterns.

The third experiment complements the second in order to generate a statistic regarding the percentage of suitable seeds in the entire seed space. Obviously, an exhaustive approach is out of the question since the seed space, which is determined by all combinations of possible values for $(a, b, x_0, y_0, z_0)$ is approximately 314 bits if computation is done in 64-bit floating point precision. Some bits are lost because valid values for all numbers in the tuple are in a small interval around 0, as stated already in [16]. Therefore, a script was developed to run the testbench for 100 seeds considering all combinations of maps and entropy builders. The parameters of the map were changed by the entropy builder at each $n$ iterations, with $n \in \{5, 10, 15, ..., 50\}$.

On top of the randomness testing, a throughput test was designed and implemented in order to evaluate the performance of the system.

*3.2. Randomness Test Results*

The entropy builder in [32] changed both parameters $a$ and $b$ by computing entropy $E_1$. In Figure 7 it can be observed that there is an obvious linear dependency between the two. This is also the case for the new entropy builder proposed in this paper, $E_2$. When using two builders concurrently, accumulating entropy differently, the parameters are completely uncorrelated and so they cover all the possible space, as depicted on the right image in Figure 7.

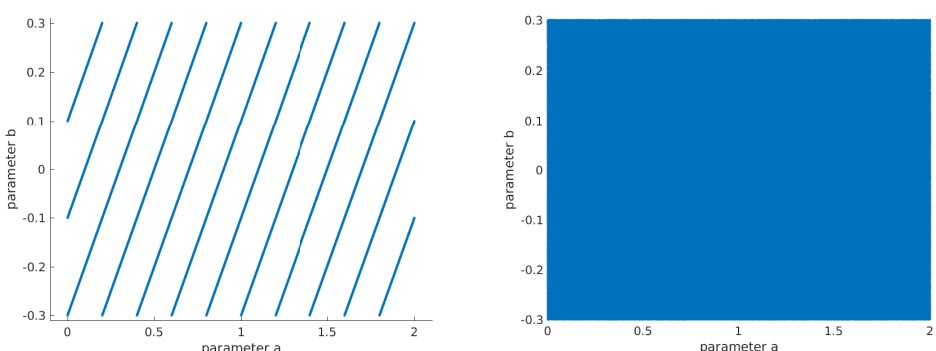

**Figure 7.** Entropy builder, $E_1$ and $E_2$ on the left, $E_1E_2$ and $E_2E_1$ on the right.

Analyzing the Lyapunov exponents for the generalized Hénon map in Figure 3, it can be seen that for $b = 0.1$ and all $a \leq 1.35$, the exponents are negative denoting pairs that do not determine chaotic behavior. Surely enough, when attempting to use them without any entropy control, all *TestU01*

and *NIST* tests fail, as it can be seen in Table 3 where several configurations were tested with the three-dimensional generalized Hénon map and a known non-chaotic initial state:

$$(a, b, x_0, y_0, z_0) = (1.35, \ 0.1, \ 0.814724, \ 0.905792, \ 0.126987)$$

When introducing an entropy builder of type $E_1$ (sum), the vast majority of tests pass. For $E_2$ builder all tests fail even when changing the parameters at each iteration. The $E_1, E_2$ combination however yields good results, as expected, passing all tests.

**Table 3.** Randomness test results for known non-chaotic state of the 3D generalized Hénon map.

| Entropy Builder | Interval | Failed TestU01 | Failed NIST |
|---|---|---|---|
| – | – | 15 | 162 |
| $E_1$ | 1 | 0 | 0 |
| $E_2$ | 1 | 15 | 162 |
| $E_1, E_2$ | 1 | 0 | 0 |
| $E_1$ | 100 | 12 | 2 |
| $E_1$ | 45 | 2 | 0 |
| $E_1$ | 35 | 0 | 0 |
| $E_1$ | 25 | 0 | 0 |
| $E_1$ | 15 | 0 | 3 |
| $E_2$ | 100 | 15 | 10 |
| $E_2$ | 45 | 3 | 1 |
| $E_2$ | 35 | 0 | 0 |
| $E_2$ | 25 | 1 | 0 |
| $E_2$ | 15 | 0 | 0 |
| $E_1, E_2$ | 100 | 14 | 4 |
| $E_1, E_2$ | 45 | 2 | 0 |
| $E_1, E_2$ | 35 | 1 | 0 |
| $E_1, E_2$ | 25 | 0 | 0 |
| $E_1, E_2$ | 15 | 0 | 0 |

For a Monte Carlo analysis, one hundred tuples were randomly selected from an uniform distribution in their corresponding definition intervals in order to characterize the generator instances. The histogram of failed tests for each generator configuration is given in Figure 8. It is clear that the correct use of an entropy builder greatly improves the randomness quality of the generator. There can also be observed that the smaller values for parameter change interval—changing the parameters more often—also improves the quality of the generator, but, as it will be seen in the next subsection, with a throughput performance penalty. Surprisingly, the lower cost sum entropy builder $E_1$ performs better than $E_2$ in terms of statistical tests results. Nevertheless, $E_2$ with an interval of 10 displays the best results of all tested configurations. $E_2$ results are the most irregular, both displaying the best and the worst results depending on the parameter change interval. One explanation would be that the *sinus* function used by $E_2$ is periodic so randomness function of the change interval is also periodic. Thus, results alternate from all tests failing for 20 and 40 to extremely good results for 10 and 30. However a combination of the two builders gives the best results on average, but still showing a performance decay with the increase of the parameter change interval.

*NF_GEN* general form in Table 2 is evaluated through several tests run with no entropy builder as well as with a generic form of the $E_1$ builder. $E_1$ sums all state values when builds up entropy and generates new parameter values by using a fractional modulo operation in order to force the values in a particular range. The considered range is $(-1; 1)$. The results for the generator without an entropy is shown in Figure 8 as the last column and it is evident that the percentage of tests passed is rather low. When using the entropy builder $E_1$, the results did not improve. On the contrary, all tests for all configurations failed. The reason, most likely, is not selecting appropriate intervals for the 10 parameters of the function. This issue needs further study in future research.

Analyzing the state-of-the-art generators selected for comparison, it is clear that even the cryptographically secure pRNG ChaCha does not pass 100% of statistical tests for all seeds tested. In fact, it scores lower that the non-CSPRNG Mersenne Twister which achieves a 95% of seeds passing all tests. What this shows is that the passing of statistical tests is a necessary but not sufficient condition for a CSPRNG. Unfortunately, we were unable to run the Monte-Carlo analysis on the Adaptive Chirikov Map, since the run-time was extremely long, and we were also unable to reproduce the results given by the authors.

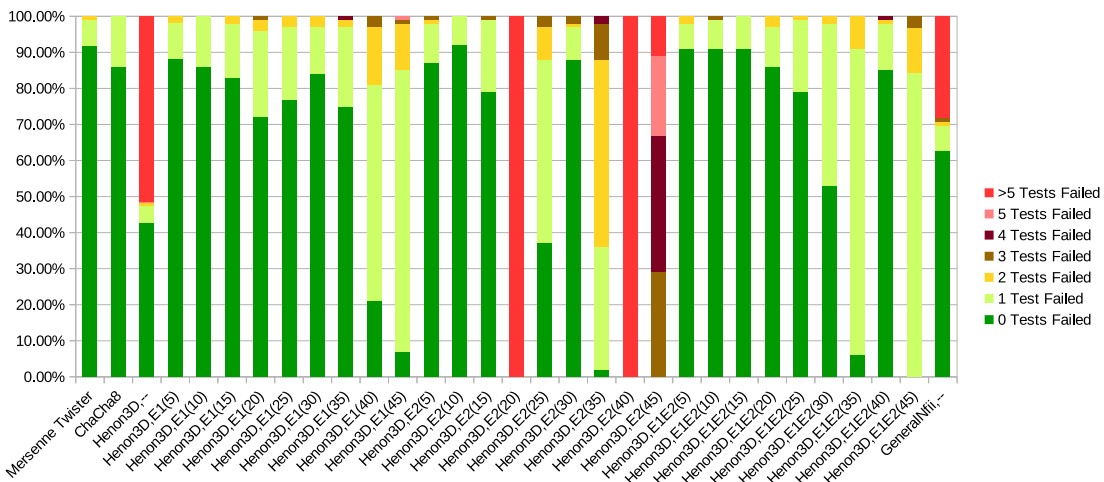

**Figure 8.** Monte Carlo analysis of seeds and failed tests for several generator configurations.

### 3.3. Throughput Test Results

The throughput analysis was done on a Lenovo 920-13IKB convertible, with an Intel Core i7-8550U processor having a maximum frequency of 4 GHz, 16 GB Dual-Channel SODIMM DDR4 at 2400 MHz, running Linux Mint 19.2 and using g++ version 7.4.0. The results are shown in Figure 9.

From the three existing algorithms, ChaCha8 is the fastest when compiled with Intel AVX extensions. This feature allows the processor to generate 32 bytes of random data at once using the 256-bit wide XMM registers. To test this, the generator function was called for each output byte (instead of generating an entire 64-byte block at a time), and as a result, the performance plummeted from 390 MB/s to 12.5 MB/s. The Mersenne Twister implementation, as well as any chaos based pRNG, can not make use of SIMD extensions, as the computation is recursive and as such, inherently sequential. As for the Adaptive Chirikov Map, because it computes four *sin* values for each random bit, it is significantly slower than all others, being able to output only 410 kB of random data each second.

Analyzing several of the proposed configurations, it can be seen that the performance of the generator depends strongly on the selected entropy builder and the parameter change interval. However, compared to the non-CSPRNG Mersenne Twister, there is a significant performance improvement when using $E_1$ as an entropy builder. The best performance is given, as expected, by the generalized Hénon Map without any entropy builder, but that configuration has a low percentage of tests passed ($\approx$50%). A good trade-off between tests passed and speed is the Hénon3D with the $E_1$ builder and 35 iterations between parameter changes. It provides an almost 300% increase of performance over the Mersenne Twister with only a 10% decrease in percentage of seeds with no test fails. A safer approach would be to use either $E_2$ with a change interval of 10 or $E_1$, $E_2$ with a change interval of 15, which pass the same amount of tests as Mersenne Twister but with about 30% increase in throughput. Compared to AVX-accelerated ChaCha8, the performance needs to be improved. This is an important target for future work and might be achieved by using several chaotic maps in parallel.

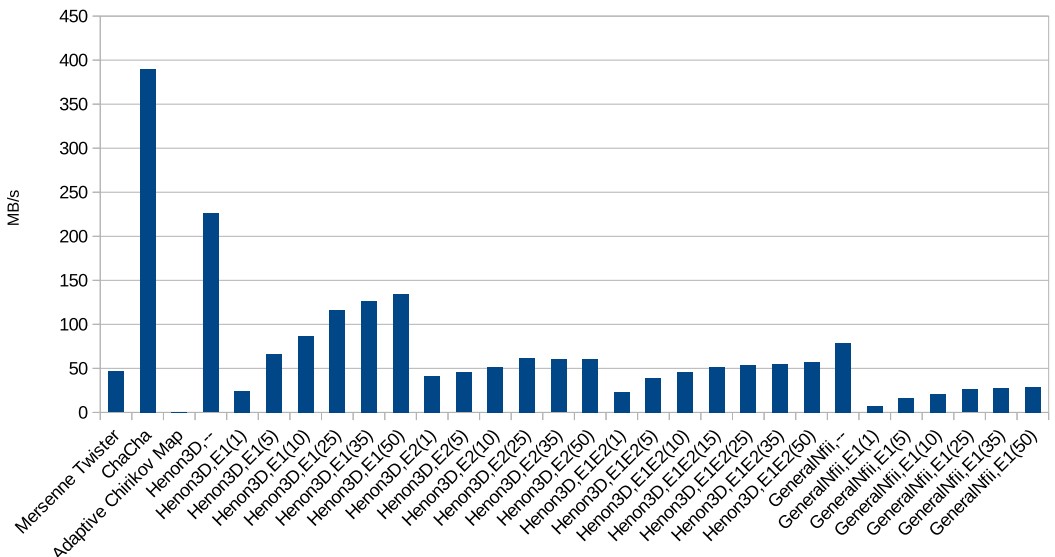

**Figure 9.** Throughput of several generator configurations.

## 4. One-Time Pad Stream Cipher Application

### 4.1. Application Design

A stream cipher is an encryption mechanism that uses a cryptographically secure pseudo-random number generator and for each byte in the clear text it generates a pseudo-random byte, producing the cipher text by *XOR-ing* the two values together. It has been proven [15] that if the generator is cryptographically secure and the same sequence is not used for multiple encryptions, the generated cipher text is impossible to discern from pseudo-random data and thus unbreakable. This section describes the implementation of such a one-time pad cipher with application on image security. The implementation uses the PNG Library in Linux to read and write images. The encryption is done on each byte on each channel on each pixel using a new pseudo-random byte. The application will write the encrypted result in the same format as the input image (grayscale, color, with or without alpha channel). The decryption is symmetric therefore it is achieved by running the same application with the encrypted image as input and the same seed used when encrypting. Running decryption with a different key yields pseudo-random data as output.

### 4.2. Results Analysis

The ideal entropy for an encrypted grayscale image is 8 [39] as it can be deduced from:

$$H(s) = -\sum_{k=0}^{2^n-1} p(k) \cdot log_2 p(k) \tag{5}$$

where $k$ is a certain gray level, $p(k)$ is the probability of occurrence of that particular gray level $k$ and $n$ is the number of bits per pixel (here $n = 8$).

An image obtained from a phone camera was enciphered in Figure 10. As it can be observed in the left side of Table 4, this plain-image (given in Figure 10, left side) already has a high entropy level. Next to it, the middle image shows the result of the encryption process when using $(a; b) = (0.15; 0.1)$ (a pair not engendering chaotic behavior for the generalized Hénon map). Its corresponding entropy is slightly greater than for the plain-image, 7.20 compared to 7.17, being still clearly visible. When $E_2$ is used alone, the right image from Figure 10 and its corresponding entropy in Table 4 show a weak encryption. The image encrypted with $(a; b) = (1.76; 0.1)$ (enabling chaos [40]), has a satisfying result of $H(s) = 7.99$, as it is the case for the entropy builder from [32] also. Switching the $a$ parameter using

$E_1$ and $b$ with $E_2$ (see (4)) gives 7.99 as well, the encryption being indiscernible, the image being similar to the one in Figure 11, bottom right.

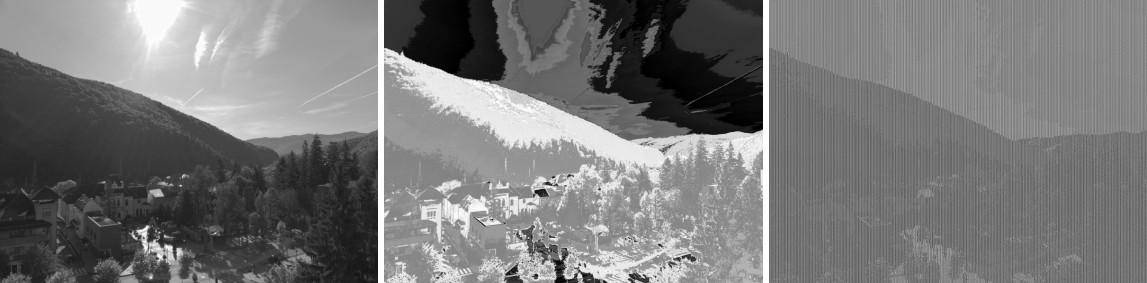

**Figure 10.** A high entropy plain-image and its poorly encrypted versions.

**Table 4.** Entropy values for plain-images in Figures 10 and 11 and their enciphered versions.

|  | Entropy |  | Entropy |
|---|---|---|---|
| original image | 7.17 | original image | 0.19 |
| encrypted; $(a; b) = (0.15; 0.1)$ | 7.20 | encrypted; $(a; b) = (0.15; 0.1)$ | 1.19 |
| encrypted; $(a; b) = (1.76; 0.1)$ | 7.99 | encrypted; $(a; b) = (1.76; 0.1)$ | 7.99 |
| encrypted with $E_1$ | 7.99 | encrypted with $E_1$ | 7.99 |
| encrypted with $E_2$ | 7.94 | encrypted with $E_2$ | 5.55 |
| encrypted using $E_1$ and $E_2$ | 7.99 | encrypted with $E_1$ and $E_2$ | 7.99 |

An image characterized by a very low entropy was used to highlight the effect of the enciphering process in Figure 11. From the left to right and top to bottom, there are shown: the plain-image, the encryption with $(a; b) = (0.15; 0.1)$, enciphering with $E_2$, and the bottom right image is the result of successful encryption when using $(a; b) = (1.76; 0.1)$, using $E_1$, respectively $E_1$ and $E_2$. The visual effects are the same as for the high level entropy plain-image in Figure 10, but the entropy computed in Table 4 is more conclusive since it shows the differences between original and cipher more clearly.

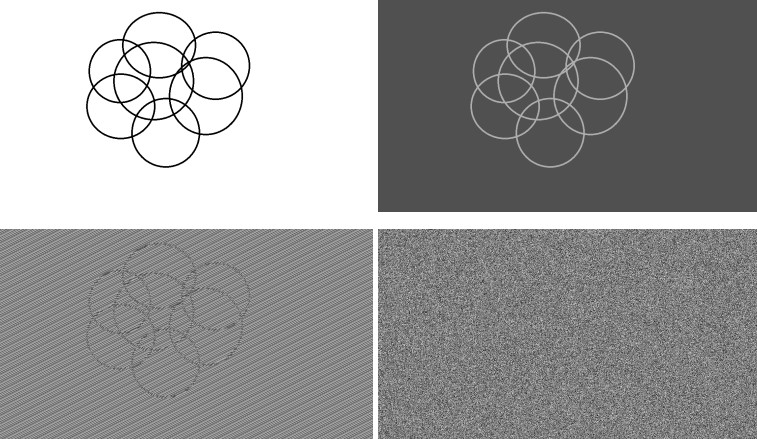

**Figure 11.** A low entropy plain-image and its encrypted versions.

Inspired by a statistical test obtaining independent and identically distributed data from chaotic maps [20] and some pRNGs using it [41,42], we enciphered the same images as in Figure 10 and in Figure 11 with different update period for the entropy builder $E_2$. Entropy values are given in Table 5 and histograms of the enciphering in Figure 12. The bottom middle histogram appears empty because the values are concentrated in 0 and 255. While update periods of 10 and 70 improve the enciphering using the entropy builder $E_2$, the distance of 20 iterations does not have the same satisfying result. Nevertheless, this is an issue that is to be thoroughly addressed in a future paper.

**Table 5.** Entropy values for encrypted images with $E_2$ and update period of 10, 20 or 70.

| Update Period | Entropy for Image in Figure 10 | Entropy for Image in Figure 11 |
|:---:|:---:|:---:|
| 10 | 7.99 | 7.99 |
| 20 | 7.17 | 0.24 |
| 70 | 7.99 | 7.99 |

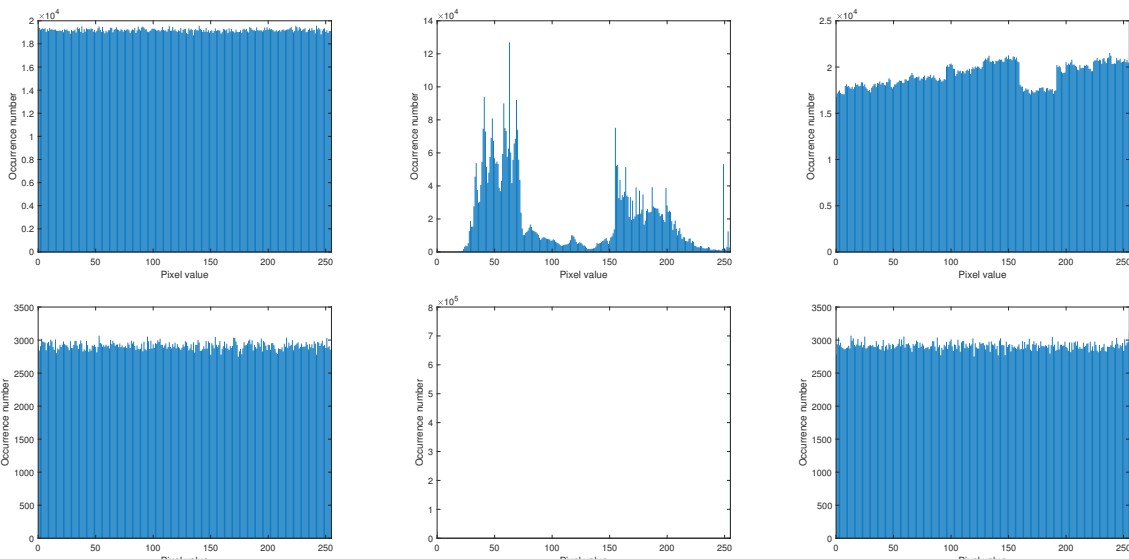

**Figure 12.** Histograms of encypherings using $E_2$ and update period of 10 (**left**), 20 (**middle**), 70 (**right**).

Further, we analyzed the correlation between adjacent pixels of the encrypted images compared to those of the plain-image. Scatter diagrams for adjacent pixels were plotted for the high entropy level plain-image in Figure 10 (top) and for the low entropy-level image in Figure 11 (bottom). Results are given in Figure 13. The top left up corner depicts the almost perfect correlation existent in the original high entropy image, while the down left corner shows the two gray levels in the low entropy chosen image. The middle correlations reflect the uniform distribution of pixel values when enciphering using the random-like behavior enabling parameters $(a; b) = (1.76; 0.1)$ or the $E_1$ entropy builder, as well as $E_1$ and $E_2$. The right up corner demonstrates a pattern in the scatter plot, when using $E_2$, indicating dependence in the values of enciphered pixels. The right bottom corner (update with $E_2$) also shows unsatisfactory results, because the two gray-levels existing in the original low entropy level image were maped to only a few different intensity levels.

Aiming to investigate the information obtained when deciphering with the wrong key, we changed each of the parameters and the initial state values at the receiving end. The computation revealed that the entropy is the same in this case as for the encrypted versions demonstrating that even the smallest change of $10^{-15}$ of any of the components of the seed will not reveal any information regarding the clear input image.

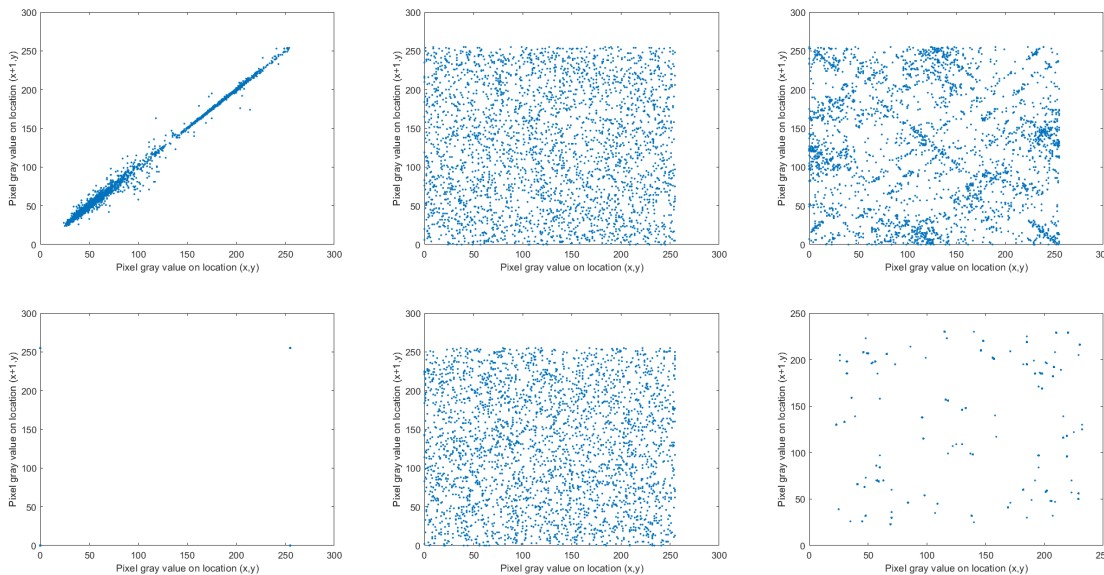

**Figure 13.** Correlation for the high entropy level image (**top**) and the low entropy level image (**bottom**).

## 5. Conclusions

In this paper the authors proposed a template scheme for a cryptographically secure pseudo-random generator based on the chaos theory. Several configurations were proposed and tested with TestU01 and NIST batteries and performance-wise against the Mersenne Twister implementation in the GNU C++ standard library, an established CSPRNG and a state of the art chaos based pRNG. The tests showed that for certain configurations, most of the seeds tested passed the statistical analysis. The throughput depends on the entropy builder function and the parameter change interval and as in most cases, a trade-off must be made between speed and security. As proof of concept, the generator was used to implement a one-time pad cryptographic system. The system was used to encipher UHD images which were then analyzed by computing entropy, through histograms and correlation coefficients. Therefore, chaotic functions are suitable building blocks for pseudo-random number generators but a mechanism for breaking patterns is necessary for ensuring randomness for all initial states. The authors plan to include the one-time pad developed in this research in a real time high resolution video streaming application. In order for this to work, throughput must be increased with a factor of 10.

**Author Contributions:** Conceptualization, O.D. and R.H.; methodology, O.D., R.H. and C.M.; software, R.H.; validation, C.M.; writing–original draft preparation, O.D.; writing–review and editing, O.D., C.M. and R.H. All authors have read and agreed to the published version of the manuscript

**Funding:** The Open Access publication cost was funded by the PubArt program of the University Politehnica of Bucharest.

**Conflicts of Interest:** The authors declare no conflict of interest. The funders had no role in the design of the study; in the collection, analyses, or interpretation of data; in the writing of the manuscript, or in the decision to publish the results.

## Abbreviations

The following abbreviations are used in this manuscript:

| | |
|---|---|
| API | Application Programming Interface |
| AVX | Advanced Vector Extension |
| CPU | Central Processing Unit |
| CSPRNG | Cryptographically Secure Pseudo-Random Number Generator |
| DDR4 | Double Data Rate (Synchronous Dynamic Random-Access Memory), version 4 |
| GCC | GNU Compiler Collection |
| GNU | GNU's Not Unix! |
| NIST | National Institute of Standards and Technology |
| PNG | Portable Network Graphics |
| pRNG | Pseudo-Random Number Generator |
| RGB | Red Green Blue |
| SIMD | Single Instruction Multiple Data |
| SODIMM | Small-Outline Dual Inline Memory Module |
| TSP | Telecommunications and Signal Processing Conference |

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
