# Peer review of "Chaos Based Cryptographic Pseudo-Random Number Generator Template with Dynamic State Changeâ€"

_applsci, doi:10.3390/app10020451_

Round 1
Reviewer 1 Report
The authors apply a chaotic map and entropy to design a PRNG which is an extension of the previous results. The manuscripts do not formulate the problem well, so that the reviewer does not capture the problem goal clearly.
Since this manuscript is an extension of the previous works, the authors should focus on the contributions of the improvements rather than stating the previous works. For example, Chaotic Map and Entropy Builder come from [2] and [1] respectively, so it is better to show the idea of the byte generator. [10] cannot be found on the Internet. Please provide clear reference, and check all others simultaneously. It is better to consider the quantification analysis to compare the proposed solution with others. For example, the authors state that the proposed solution generates more than 90% results which are different to the initial state. How is the 90% captured? Is 90% better than the state-of-the-art result? Can the randomness be formulated? If yes, please state the objective function. If no, the reviewer believes that the problem is not clear enough. P5 line 90, The authors stated that "A general form for a 3D map is given in (4)." However, (4) is the entropy in p 10. Is that corrected? The manuscript lacks for the discussion of the advantage of the design. For example, why consider the byte generator, and what is the advantage for the design as shown in Figure 1? It should be well stated, and the readers are able to capture the critical idea of the approach design.Author Response
Dear reviewer, thank you very much for taking the time to review our paper! Please see our responses below:
The authors apply a chaotic map and entropy to design a PRNG which is an extension of the previous results. The manuscripts do not formulate the problem well, so that the reviewer does not capture the problem goal clearly.
We have attempted to clarify the goal of the paper and specify the addressed problem more clearly. (lines 20-36).
Since this manuscript is an extension of the previous works, the authors should focus on the contributions of the improvements rather than stating the previous works. For example, Chaotic Map and Entropy Builder come from [2] and [1] respectively, so it is better to show the idea of the byte generator.
We have added detail regarding the novelty of our proposal (lines 92-100).
[10] cannot be found on the Internet. Please provide clear reference, and check all others simultaneously.
We have added the on-line reference for [10]. We also checked the other references. Thank you!
It is better to consider the quantification analysis to compare the proposed solution with others. For example, the authors state that the proposed solution generates more than 90% results which are different to the initial state. How is the 90% captured? Is 90% better than the state-of-the-art result? Can the randomness be formulated? If yes, please state the objective function. If no, the reviewer believes that the problem is not clear enough.
We have clarified in the abstract that the 90% is obtained by a Monte Carlo analysis of the seed space, randomly selecting 100 seeds and running the statistical tests. Also, lines 195-196 and the label of Fig. 8 was updated to reflect that change. State-of-the-art results are discussed on lines 217-223.
The results for the empirical statistical tests were used as an objective function, comparing the p-value with a level of the statistical significance alpha = 0.1. This is specified in the documentation of the statistical tests, If you believe we should add this in our paper, please let us know.
P5 line 90, The authors stated that "A general form for a 3D map is given in (4)." However, (4) is the entropy in p 10. Is that corrected?
We have corrected the number referring to the system. New line is 120. Thank you!
The manuscript lacks for the discussion of the advantage of the design. For example, why consider the byte generator, and what is the advantage for the design as shown in Figure 1? It should be well stated, and the readers are able to capture the critical idea of the approach design.
We have added additional clarifications regarding the flexibility of the design on lines 108-114.
Reviewer 2 Report
Dear Authors!
It was interesting to me to review your manuscript containing such a thorough technical study. The chaos-based encryption is a modern direction in cryptography. However, I could not recommend your article for publication in present form due the following reasons:
1. Lines 20, 24, 30 and some others contain an error. The correct number for the Hénon generalized map is (1) instead of (2).
2. In line 22, you specify the map parameters. However, study [3] examined a wider range of parameters. Why does your study considered a narrow interval?
3. It is better to move the values from table 1 to the caption of Figure 2 and add the notation a, b, etc.
4. You call the investigated mapas a the Henon map, that is incorrect. The original Henon map consists of two mathematical expressions. You are considering its modification called the Hénon generalized map. Please check it.
5. It is not clear which algorithm you mention on line 31 (The algorithm sorts the exponents from the largest to the lowest). It is worth clarifying that we are talking about a method for calculating the Lyapunov spectrum.
6. To ensure that changing the map parameters does not lead to periodic behavior, you need to plot a two-dimensional bifurcation diagram in the intervals that are mentioned in formula (2). This should be done in order to exclude the pairs of parameters that are not suitable for the pseudo-random number generation. Then, your entropy calculation experiments will be appropriate. The methods for constructing two-dimensional bifurcation diagrams are described in detail in the article by D.N.Bususov et al. Comparing the algorithms of multiparametric bifurcation analysis (DOI: 10.1109/SCM.2017.7970536)
7. Perhaps in line 114 it is more correct to use “pseudo-random” instead of “random”.
8. A review of recent advances in chaotic cryptography should be performed. Some new results are presented in articles by E.G. Nepomuceno (Image encryption based on the pseudo-orbits from 1D chaotic map), Pesterev D.O. (1. Novel normalization technique for chaotic Pseudo-random number generators based on semi-implicit ODE solvers 2. Adaptive Chirikov Map for Pseudo-random Number Generation in Chaos-based Stream Encryption). Please review these and other studies and expand the state-of-the-art section and the refences list.
9. It is of interest to compare the encryption using the proposed method with traditional algorithms including AES since the main advantage of chaotic encryption is speed. Have you performed a performance assessment not only for the generator, but also for the ecryption sheme as a whole?
Thus, my decision is the major revision.
Author Response
Dear reviewer, thank you very much for taking the time to review our paper! Please see our responses below:
1. Lines 20, 24, 30 and some others contain an error. The correct number for the Hénon generalized map is (1) instead of (2).
We corrected the number referring to the generalized Henon map. Thank you!
2. In line 22, you specify the map parameters. However, study [3] examined a wider range of parameters. Why does your study considered a narrow interval?
We added a detail about the ranges for parameters a and b on lines 41-45.
3. It is better to move the values from table 1 to the caption of Figure 2 and add the notation a, b, etc.
Unfortunately, the template instructions specifically limited the figure and table captions to one line. This is why we chose to insert the values as a table. The only other option would be to simply embed them in the text. Please advise!
4. You call the investigated mapas a the Henon map, that is incorrect. The original Henon map consists of two mathematical expressions. You are considering its modification called the Hénon generalized map. Please check it.
We have corrected the name of the system, in the text. Thank you!
5. It is not clear which algorithm you mention on line 31 (The algorithm sorts the exponents from the largest to the lowest). It is worth clarifying that we are talking about a method for calculating the Lyapunov spectrum.
Corrected. Thank you! (lines 58-59)
6. To ensure that changing the map parameters does not lead to periodic behavior, you need to plot a two-dimensional bifurcation diagram in the intervals that are mentioned in formula (2). This should be done in order to exclude the pairs of parameters that are not suitable for the pseudo-random number generation. Then, your entropy calculation experiments will be appropriate. The methods for constructing two-dimensional bifurcation diagrams are described in detail in the article by D.N.Bususov et al. Comparing the algorithms of multiparametric bifurcation analysis (DOI: 10.1109/SCM.2017.7970536)
Avoiding periodic behavior is one of the goals of the paper which we solve by periodically perturbing the map's parameters using the entropy information gathered by the builder. Therefore, we feel that an analysis of this kind is not necessary. If this is not clear enough, let us know, we will attempt to add more detail.
7. Perhaps in line 114 it is more correct to use “pseudo-random” instead of “random”.
Corrected. Thank you!
8. A review of recent advances in chaotic cryptography should be performed. Some new results are presented in articles by E.G. Nepomuceno (Image encryption based on the pseudo-orbits from 1D chaotic map), Pesterev D.O. (1. Novel normalization technique for chaotic Pseudo-random number generators based on semi-implicit ODE solvers 2. Adaptive Chirikov Map for Pseudo-random Number Generation in Chaos-based Stream Encryption). Please review these and other studies and expand the state-of-the-art section and the refences list.
We have updated the paper with the provided titles and some other recent research. Thank you for the ones you provided!
9. It is of interest to compare the encryption using the proposed method with traditional algorithms including AES since the main advantage of chaotic encryption is speed. Have you performed a performance assessment not only for the generator, but also for the ecryption sheme as a whole?
Since the main object of our paper is the pRNG, we have only compared the randomness and throughput to another well established pRNGs - Mersenne Twister, ChaCha8 and Adaptive Chirikov Map (the latter two added after your review). The one-time pad implementation is given only as a possible application to the generator, and such we didn't believe that a comparison would benefit our paper. Moreover, AES is a block-cipher and as such, it has several blocking modes, with different speed performances. Finally, AES, being the current standard for symmetric encryption, has hardware acceleration support in most Intel processors. So in these circumstances, we concluded that a comparison would not be fair. However, in order to broaden the comparisons, we have added the two new random generators mentioned above to the comparison table.
Round 2
Reviewer 2 Report
Dear Authors,
Thank you for taking my recommendations into account.
Despite the fact that the focus of the article is on the PRNG, you mention an encryption scheme based on the proposed generator. Therefore, comparison with any traditional and widely-used schemes is of interest. I understand that such study requires additional experiments, so I propose to address this topic in your follow-up article.
Thus, my decision is accept in present form.